# Carbon Atoms Speaking Out: How the Geometric Sensitivity of ^13^C Chemical Shifts Leads to Understanding the Colour Tuning of Phycocyanobilin in Cph1 and AnPixJ

**DOI:** 10.3390/molecules25235505

**Published:** 2020-11-24

**Authors:** Sascha Jähnigen, Daniel Sebastiani

**Affiliations:** Institut für Chemie, Naturwissenschaftliche Fakultät II, Martin-Luther-Universität Halle-Wittenberg, von-Danckelmann-Platz 4, 06120 Halle (Saale), Germany; sascha.jahnigen@ens.psl.eu

**Keywords:** protein NMR, theoretical spectroscopy, phycocyanobilin, π-conjugation, colour tuning, multivariate statistics, QM/MM, molecular dynamics

## Abstract

We present a combined quantum mechanics/molecular mechanics (QM/MM) molecular dynamics–statistical approach for the interpretation of nuclear magnetic resonance (NMR) chemical shift patterns in phycocyanobilin (PCB). These were originally associated with colour tuning upon photoproduct formation in red/green-absorbing cyanobacteriochrome AnPixJg2 and red/far-red-absorbing phytochrome Cph1Δ2. We pursue an indirect approach without computation of the absorption frequencies since the molecular geometry of cofactor and protein are not accurately known. Instead, we resort to a heuristic determination of the conjugation length in PCB through the experimental NMR chemical shift patterns, supported by quantum chemical calculations. We have found a characteristic correlation pattern of ${}^{13}$C chemical shifts to specific bond orders within the π-conjugated system, which rests on the relative position of carbon atoms with respect to electron-withdrawing groups and the polarisation of covalent bonds. We propose the inversion of this regioselective relationship using multivariate statistics and to apply it to the known experimental NMR chemical shifts in order to predict changes in the bond alternation pattern. Therefrom the extent of electronic conjugation, and eventually the change in absorption frequency, can be derived. In the process, the consultation of explicit mesomeric formulae plays an important role to qualitatively account for possible conjugation scenarios of the chromophore. While we are able to consistently associate the NMR chemical shifts with hypsochromic and bathochromic shifts in the P${}_{g}$ and P${}_{\mathrm{fr}}$, our approach represents an alternative method to increase the explanatory power of NMR spectroscopic data in proteins.

## 1. Introduction

Tetrapyrroles showcase an overwhelming abundance in living species, accompanied by an impressive functional versatility [1]. They can be classified into cyclic and open-chain forms, the former comprising the well-known porphyrins that build the precursors of hemes and chlorophylls, renowned for their important role in oxygen transport, oxidoreductase enzymes, and photosynthesis [2]. The second group, open-chain tetrapyrroles, also referred to as *bilins*, are ubiquitous in living species as well. Organised in biliproteins, they take the role of a chromophore and serve phototrophic organisms as accessory pigments [3]. An important class of biliproteins is formed by phytochromes—photosensors that typically entail histidine kinase activity to regulate photomorphogenesis and photoperiodism [4,5,6,7]. The occurrence of phytochromes was long assumed to be restricted to higher plants, but over the years members of this class have also been found in cyanobacteria, bacteria, and fungi [6,7,8,9,10,11]. Recently, variants of (bacterio)phytochromes were found, genuine to cyanobacteria, that are therefore called cyanobacteriochromes (CBCRs) [12,13]. They fulfil a widespread range of tasks in the cell connected to the response towards light, such as positive phototaxis [12,14,15]. Although being phylogenetically related to phytochromes, CBCRs bear unique features: while the sensory module of phytochromes needs to encompass at least a period circadian/aryl hydrocarbon receptor nuclear translocator/single-minded (PAS)–cGMP phosphodiesterase/adenylyl cyclase/FhlA (GAF)–phytochrome specific (PHY) tridomain (Figure 1), CBCRs require only a chromophore-binding GAF domain to maintain their biochemical properties [16]. Most intriguing, however, is the ability of CBCRs to tune their absorption over the entire range of the visible spectrum, including near-IR and near-UV [12,17,18,19,20]. The treasure of photosensors provided by nature with which organisms adapt to their environment bears high utility for biochemical applications in bioimaging and optogenetics, but also in therapy and catalysis [14,21,22,23,24,25,26,27,28,29,30,31,32,33,34].

Many cyanobacterial phytochromes and CBCRs carry as bilin cofactor, phycocyanobilin (PCB), which operates as a photoreversible switch, toggling between the *ZZZssa* and *ZZEssa* configuration (Scheme 1) [4]. Exciting the chromophore at its absorption maximum, photochemical *Z*/*E* isomerisation of the C${}_{15}$–C${}_{16}$ double bond initiates a flip of the D ring, which triggers a biochemical response [6,36,37,38,39,40,41]. The two states, *Z* and *E*, are denoted as dark state and photoproduct, respectively. The photoproduct is back-transformed into the dark state by photochemical *E*/*Z* isomerisation after excitation at its, typically shifted, absorption maximum. Depending on the colour (*x*) of absorbed light, the states are labelled as “P${}_{x}$”. Phytochromes switch between red-absorbing (P${}_{r}$) dark states and far-red-absorbing (P${}_{\mathrm{fr}}$) photoproducts, [4,8] while the colour diversity in CBCRs is broader. Typical examples of CBCR absorption colour pairs are red/green, green/red, blue/green, red/orange, green/orange, violet/orange, blue/yellow, or blue/teal [12,14,18,19,20,34,42].

The colour tuning in CBCRs is exceptional, regarding the very limited number of chromophore species they bind. We recently showed that the degree of solvation due to a sealed chromophore binding pocket cannot serve as an explanation model [43]. However, Peng and co-workers experimentally related the dihedral angle of ring D of PCB in allophycocyanin B (a phycobilisome) with the absorption maxima ranging from 590 to 670 nm and thus proved the bathochromic shift’s originating in an increased conjugation length [44]. Recently, Wiebeler and co-workers showed the hypsochromic shift in the green-absorbing photoproduct of Slr1393g3 (a PCB-carrying CBCR) to emerge from a decrease of conjugation, originating from the out-of-plane rotation of ring D (“trapped-twist” model) [17,45]. They underlined the important role of the apoprotein to keep the chromophore in the intended conformational state.

In this work, we present a quantum mechanics/molecular mechanics (QM/MM) study of PCB in a protein environment by means of extensive molecular dynamics (MD) and ab initio ${}^{13}$C-nuclear magnetic resonance (NMR) chemical shift calculations, based on density functional theory (DFT). Comprising the entire protein together with the solvent, our theoretical model is able to account for physical meaningful non-covalent interactions without losing accuracy at describing the chromophore and its instantaneous thermal fluctuations. We focus on two cyanobacterial photosensor proteins (Figure 1): the red/far-red switching Cph1Δ2 (see “Abbreviations”), a well-studied member of the phytochrome superfamily [10,18,35,38,39,40,46,47,48,49,50,51,52,53,54,55,56,57,58,59,60,61,62,63,64,65,66] and the red/green switching AnPixJg2 (see “Abbreviations”), a cyanobacteriochrome that has been discovered later [12,13,43,45,67,68,69,70,71,72,73]. Both proteins have PCB as a cofactor and exhibit a red-absorbing dark state (P${}_{r}$). Photoconversion yields a far-red absorbing photoproduct (P${}_{\mathrm{fr}}$) in the case of Cph1Δ2, whereas AnPixJg2 renders a green-absorbing state (P${}_{g}$). Although the spectral shift in opposite directions amounts to a difference of more than 100 nm, photoproduct formation neither implies chemical changes to the chromophore nor a changed protonation state [45,70,72]. However, according to the studies by Peng and Wiebeler it can be concluded that the D-ring twist and thus the effective conjugation length determine the absorption properties [17,44]. To date, no crystal structures of the photoproduct states of Cph1Δ2 and AnPixJg2 exist that could build the basis of theoretical structure models, but we show that relying on the two P${}_{r}$ states, structural fluctuations can profitably be correlated with calculated spectroscopic patterns and increase the explanatory power of NMR spectroscopy towards conjugation effects in the P${}_{g}$ and P${}_{\mathrm{fr}}$ state, respectively.

## 2. Results

### 2.1. How Bond Alternation and Thermal Fluctuations Determine ^13^C-NMR Chemical Shifts

Extensive MD sampling under the QM/MM regime allows for the inclusion of thermal fluctuations of the molecular structure on the femtosecond and picosecond time scale. Figure 2a shows the computed time evolution of the C${}_{13}$–C${}_{14}$ bond length and the ${}^{13}$C chemical shift of atoms C${}_{13}$ and C${}_{14}$—exemplary for all similar features in PCB—as they are found in the simulations. The thermal motion of the atoms at 300 K leads to bond length oscillations in the range 1.4–1.5 Å, and to corresponding oscillations of the instantaneous carbon NMR chemical shifts within a range of about 20 ppm. These oscillations are too fast for the NMR experiment; only their averages (dashed lines in Figure 2a, cf. also Table 1) can be compared to corresponding measurements. Figure 2b matches computed ${}^{13}$C chemical shifts of PCB’s carbon atom scaffold (C${}_{1}$ through C${}_{19}$) with reported MAS-NMR data from Matysik and co-workers for the P${}_{r}$ states of AnPixJg2 and Cph1Δ2 at 233 K [68]. The simulations clearly reproduce the characteristic alternating pattern of ${}^{13}$C chemical shifts mirroring the π-conjugated system, which only proves that PCB in the model represents the physical state of the chromophore in the proteins. At the same time, though, the uncertainty in the computed averages impedes distinguishing between the two protein environments of the cofactor. In fact, with chemical shifts differing by less than 5 ppm, AnPixJg2 and Cph1Δ2 are hardly distinguishable experimentally let alone by the QM/MM model given the large fluctuations shown in Figure 2a.

Although it seems that the uncertainty in the obtained averages of ${}^{13}$C chemical shifts suggests a high sensitivity of the carbon atoms’ shielding towards any direct or indirect change in the environment, the further analysis of their dependence on chromophore–apoprotein connections and temperature effects enables a more refined notion of the matter. Computationally, it is straightforward to repeat chemical shift calculations for the same setup, gradually “switching-on” those protein residues that interact with the chromophore, or numerically “cooling-down” the system, which forces the atoms back to their equilibrium positions. Corresponding results can be found in Appendix A (Figure A1). These technical modifications shed light into the susceptibility of ${}^{13}$C chemical shifts in PCB—and into those of other nuclei. It turns out that they are largely insensitive to temperature and non-covalent interactions the chromophore is subjected to due to their mainly quartary nature. This contrasts with the high sensitivity of ${}^{1}$H and ${}^{15}$N nuclei, the chemical shifts of which change strongly with both temperature and supramolecular interactions.

However, Figure 2b undoubtedly indicates that the conjugation pattern and charge distribution in the chromophore strongly determine the extent of ${}^{13}$C chemical shifts. This striking correspondence points to the fact that instantaneous fluctuations like those shown in Figure 2a actually incorporate an intimate connection of bond parameters and nuclear shieldings. How this manifests in PCB will be shown and discussed in the following section; we so far conclude that the chromophore’s carbon atoms C${}_{1}$–C${}_{19}$, though being blind towards supramolecular effects, are an important probe of their immediate covalent situation.

### 2.2. Correlations of ^13^C Chemical Shifts and Their Geometric Sensitivity

The interconnection of nuclear shieldings and bond lengths is not particularly surprising, but the remarkable “isolation” of ${}^{13}$C chemical shifts from supramolecular interactions deserves a closer look. Figure 3 presents the linear correlation matrix of ${}^{13}$C shifts with C-C bond lengths in rings C and D of PCB in P${}_{r}$ state. It is based on plotting the instantaneous values obtained from the QM/MM trajectories against each other and extracting statistically significant connections through linear regression. It therefore reveals whether chemical shifts decrease or increase upon increasing the individual bond lengths (see Figure A2 for the explicit correlation diagram). It is important to note that we are entitled to carry out a combined analysis of both proteins alike, given the P${}_{r}$ state of both proteins, AnPixJg2 and Cph1Δ2, having PCB in the same or a very similar configuration (the matrix shown in Figure 3 looks essentially the same for either protein). It is the central result of this report, because it identifies not only pivotal correlations, but furthermore reveals a particular pattern, which in the following we refer to as the geometric sensitivity of ${}^{13}$C chemical shifts. There are numerous representations that describe a sensitivity, yet in this study we are interested in those chromophore parts that are involved in colour tuning, that is, the nuclei describing the conjugation between rings C and D (C${}_{13}$–C${}_{19}$).

Qualitatively, the geometric sensitivity follows the clear rationale that an atom’s chemical shift increases with the length of the originating bonds—and decreases with the next, and increases again with the following bond and so on. In Figure 3 this alternation is marked by red and blue bullets, respectively, shaping an imagined double diagonal; like Figure 2b, it is another representation of the underlying conjugated system. The symmetry of the geometric sensitivity, however, is clearly disturbed leaving certain nuclei with a “preferred” bond they are correlated with. For instance, there is a strong connection of atoms C${}_{13}$ and C${}_{14}$ to bond C${}_{13}$–C${}_{14}$, but a rather weak one to bonds C${}_{12}$–C${}_{13}$ and C${}_{14}$–C${}_{15}$, respectively. The same observation can be made for C${}_{16}$ and C${}_{17}$, while the latter shows a strong connection to bond C${}_{18}$–C${}_{19}$, whereof it is not even a member. Expectedly, the two pyrrole systems mainly exhibit within-ring correlations, but atoms C${}_{15}$ and C${}_{16}$ are connected to both parts. Amide C${}_{19}$, in turn, appears completely independent from the conjugated system.

The characteristic sensitivity in Figure 3 evidently is determined not only by the bond alternation of the conjugated system, but also by bond polarisation due to the relative position of the nuclei to electron-withdrawing groups and substituents. Nevertheless, the direct interpretation of the geometric pattern is difficult and we suggest employing tools from multivariate statistics. As will be shown in the following section, we can use the geometric sensitivity together with experimental evidence to predict the conjugation length in P${}_{g}$ and P${}_{\mathrm{fr}}$ states of PCB alike, assuming that the correlation matrix in Figure 3 remains unaltered in either state.

### 2.3. Inversion of the Geometric Sensitivity Points to P_g_ and P_fr_ States

In 2015, Song et al. reported a peculiar difference pattern of ${}^{13}$C nuclear shieldings (Δδ${}^{c}$) in PCB upon photoconversion in AnPixJg2 and Cph1Δ2: Nuclei belonging to rings C and D exhibit changes in chemical shifts of up to 10 ppm when transforming from the P${}_{r}$ state into the respective photoproduct [68]. The authors attribute this observation to a “large-scale rearrangement of the critical interactions [that] cannot be explained only by the configurational changes of the chromophore.” However, recalling that ${}^{13}$C chemical shifts are blind towards supramolecular perturbations (cf. Figure A1) the changes cannot be explained by the removal of hydrogen bonds or the emersion of a new charged side chain—unless this involves changing the covalent situation of the carbon atoms. Most intriguing, however, is the observation by Song of an opposite trend of the Δδ${}^{c}$ pattern for most atoms upon switching to either P${}_{\mathrm{fr}}$ or P${}_{g}$ state with P${}_{r}$ lying in the middle. Following the correspondence principle, this trend must be found in the causing process (i.e., photoconversion) as well, which is supported by what is by now commonly understood as the colour tuning mechanism: [17,44] the extent of conjugation in rings B, C and D of the chromophore, where the P${}_{g}$ and P${}_{\mathrm{fr}}$ states are characterised by a decreasing and increasing conjugation length relative to the P${}_{r}$ state, respectively.

The NMR experimental observations are made amid scarce information on the atomic coordinates of the photoproduct states since no structural data from X-ray diffraction are available. In these terms, the geometric sensitivity may serve to retrieve structural data from the ${}^{13}$C-NMR experiment and to predict the single–double bond conjugation pattern for the P${}_{g}$ and the P${}_{\mathrm{fr}}$ state. To endow our theoretical observations with quantitative information, we inverted the correlation matrix in Figure 3 and applied the result on the Δδ${}^{c}$ patterns reported by Song et al. [68] We carried out a principal component analysis (PCA) to account for the importance of fluctuations in the feature variables before using multivariate regression to obtain predicted Δd patterns that correspond to the experimental Δδ${}^{c}$ patterns, as shown in Table 1 (see Appendix A for computational details). We report as a reference the average bond lengths of PCB in the P${}_{r}$ state, calculated from the MD trajectories. They support the inferred change in bond order, for instance to what extent a single or double bond character is retained. The given uncertainties of prediction help to distinguish important from less significant trends upon photoconversion.

Comparing the two P${}_{r}$ states of AnPixJg2 and Cph1Δ2, we do not find big differences between the two proteins, which corresponds to the earlier argument that PCB in both proteins is chemically equivalent. The found numbers are mainly due to the comparably large difference in chemical shift for atom C${}_{15}$, lying at the joint of rings C and D and which is connected to many bonds (cf. Figure 3). Going over to predicted changes in bond length upon photoconversion, that is, differences going from P${}_{r}$ to P${}_{g}$ and P${}_{\mathrm{fr}}$, respectively, we are able to distinguish key sites among other, less significant changes. Bond C${}_{13}$–C${}_{14}$, for which we hardly see a difference in the P${}_{r}$ state, undergoes important changes and exhibits a clearly opposite behaviour in the process: while in P${}_{g}$ state it is shortened by about 17 pm, its length increases by about 13 pm in P${}_{\mathrm{fr}}$. Bond C${}_{14}$–C${}_{15}$ is already longer in P${}_{r}$ of AnPixJg2 and becomes even longer in P${}_{g}$, but shorter in P${}_{\mathrm{fr}}$ of Cph1Δ2, amounting to a net difference of 12 pm between the photoproducts. Bond C${}_{15}$–C${}_{16}$ again shows opposite behaviour being shorter in the P${}_{g}$ state (–14 pm) and longer in the P${}_{\mathrm{fr}}$ state (+8 pm). For the bonds forming ring D the prediction tends to assign changes to either one or the other photoproduct, but changes in bond C${}_{16}$–C${}_{17}$ are small and not significant. Bond C${}_{17}$–C${}_{18}$ is clearly lengthened in P${}_{\mathrm{fr}}$ (+19 pm) as is bond C${}_{18}$–C${}_{19}$ in P${}_{g}$ (+16 pm) with respect to P${}_{r}$. Consequently, this analysis reveals that it is possible turning the asymmetry of the geometric sensitivity of ${}^{13}$C chemical shifts to precisely account for subtle changes in the covalent structure of PCB, thereby allowing regioselective predictions. In the following discussion we make the attempt to interpret the pattern on a molecular scale.

## 3. Discussion

The presented results of the inversion of the geometric sensitivity of ${}^{13}$C chemical shifts are based on experimental evidence reported by Song et al. [68] We inferred expected changes in bond length of the PCB chromophore when going from the dark state P${}_{r}$ to the photostates P${}_{g}$ for AnPixJg2 and P${}_{\mathrm{fr}}$ for Cph1Δ2. We pointed out—in accordance with the common understanding of the colour tuning mechanism in these forms—the antagonistic nature of the changes involved. The results in Table 1 can be used to predict the predominant character of the investigated C-C bonds of PCB in P${}_{g}$ and P${}_{\mathrm{fr}}$ states. Associating bond lengths with bond orders, we propose the following alternation pattern:

Table 2 suggests that the bond alternation pattern in the two photostates is opposite and we present in the following an explanatory model to translate this into the effective conjugation length. One has to keep in mind that the alternation pattern represents the predominant character of the C-C bonds, but does not compare to pure single or double bonds since pyrrole rings form a π-conjugated system not only between, but also within themselves.

Considering that the shaping of delocalisation of double bonds by means of mesomeric schemes has so far not received due attention, we argue that working out what is actually possible and favourable in terms of bond alternation is a key to fully understand the colour tuning phenomenon in PCB.

### 3.1. P_g_: Short Conjugation

Scheme 2 and Scheme 3 illustrate the π-resonance of PCB in P${}_{g}$ state where conjugation has been found to be of short length. We consider the conjugation between rings B and C, which is maintained in all states of the chromophore, as *core conjugation*. Assuming that in P${}_{g}$ state ring D is twisted out of plane, core conjugation alone carries the absorption of green light, which is in line with what has been reported by Wiebeler et al. [17] There exist several mesomeric formulae describing this conjugation, the most important of which are shown in Scheme 2. Therein, the positive charge is delocalised along with the change of bond alternation, but the high electronegativity of the nitrogen atoms shifts the distribution towards the formation of stabilised tertiary carbocations centered at C${}_{8}$ and C${}_{12}$, thereby rendering bond C${}_{13}$–C${}_{14}$ with an increased bond order. A stabilisation of the positive charge to some extent at the pyrrole nitrogen atoms seems reasonable since both proteins present residues at this location that may serve as hydrogen bond acceptors, but have to compete with water molecules—also in AnPixJg2 [43]. In any case this would not swivel the predicted bond character at C${}_{13}$–C${}_{14}$.

Simultaneously, ring D, being fully detached from the conjugated core, is subjected to *amide resonance* pulling electron density from the pyrrole ring and generating a negative partial charge at the carbonyl oxygen atom. However, in pyrrole, resonance does not only include the NH group of the amide bond, but the entire aromatic system. As a consequence another mesomeric form can be found, which is again stabilised through formation of a tertiary carbocation far from the electronegative centres. This has the important consequence that bond C${}_{17}$–C${}_{18}$ no longer retains a double bond character, which is shifted to C${}_{18}$–C${}_{19}$, but that of a single bond with atom C${}_{17}$ carrying the positive (partial) charge.

We want to note that the methylene bridges are not favourable sites to put a positive partial charge, due to their secondary type, but they still take part in conjugation. Furthermore, it has been argued that a certain single bond character of C${}_{15}$–C${}_{16}$ is the underlying cause of “dark conversion” [6].

We conclude that a plausible prediction of the geometric bond length conjugation pattern of the PCB in its P${}_{g}$ state is possible through measured and computed NMR chemical shifts (and their geometric sensitivities). The resulting conjugation pattern, in turn, provides a consistent explanation of the change in absorption wavelength after photoconversion, which is an explanation for the unusual colour change of the bilin chromophore in this protein.

### 3.2. P_fr_: Long Conjugation

π-Resonance of PCB in P${}_{\mathrm{fr}}$ state follows different principles than in the case for the P${}_{g}$ state. As was pointed out by Peng et al., the increase in absorption wavelength is due to the extension of conjugation, which now includes rings B, C, and D since the latter rotates in the plane with the core [44]. We shall call this *full conjugation* because it resembles the longest possible resonance in PCB. Including ring D in the exchange of π-electrons has not only bonds C${}_{14}$–C${}_{15}$ and C${}_{15}$–C${}_{16}$ toggle their single–double bond pattern, but also withdraws charge density from the ring as the positive charge is now shared as well.

Scheme 4 presents the most important mesomeric formulae in the P${}_{\mathrm{fr}}$ state and an immediate question might be why there is no favourable carbocation formation in ring D as has been adduced for rings B and C. The reason behind this is that in the case of P${}_{\mathrm{fr}}$, an explicit interaction with the apoprotein makes a difference. We have argued before that the carbon atoms do not directly “see” supramolecular effects; however, they are sensitive to indirect effects, which in the present case is the stabilisation of the positive charge by the apoprotein: Rohmer et al. argued that in P${}_{\mathrm{fr}}$ the negatively charged Asp207 is in position to form a salt bridge with the pyrrole nitrogen atom carrying the positive charge [38]. Based on our analysis, we strongly suggest that this interaction is the dominating factor in the π-resonance of the P${}_{\mathrm{fr}}$ state. The new stable mesomeric form that can be found in the P${}_{\mathrm{fr}}$ state corresponds with the predicted bond alternation pattern in Table 2.

In summary, also for PCB in its P${}_{\mathrm{fr}}$ state a plausible prediction of the bond length conjugation pattern is obtained, based solely on measured and computed NMR chemical shifts (and their geometric sensitivities). Again, the resulting conjugation pattern serves as explanation of the change in absorption wavelength through photoconversion.

## 4. Conclusions

We have addressed the colour tuning mechanism in the photosensor proteins AnPixJg2 and Cph1Δ2, which are based on the phycocyanobilin (PCB) chromophore, via a quantum chemical method using an indirect approach. While the direct approach (i.e., the calculation of absorption energies in the different configurations) is possible, but difficult due to the absence of reliable structural data, we have chosen to use reported experimental ${}^{13}$C-NMR chemical shifts of the chromophore backbone. These data are available for both P${}_{\mathrm{fr}}$ and P${}_{g}$ state, together with the chemical shift changes with respect to the dark state conformation, P${}_{r}$, of known structure. We infer structural information and derive the respective bond length alternation pattern, which points towards the effective conjugation in the unknown P${}_{\mathrm{fr}}$ and P${}_{g}$ states. This can eventually be linked to the changes in absorption wavelength via simple confinement considerations.

Our complementary approach represents an uncommon computational route that is interesting for situations in which clear and reliable structural information (e.g., from X-ray diffraction) is difficult to obtain due to experimental constraints (here, the lifetime of the photoproducts P${}_{\mathrm{fr}}$ and P${}_{g}$). The approach requires the quantitative determination and subsequent inversion of a particular structure–property relationship; namely, the dependence of the NMR chemical shifts on the bond length pattern. This can be realised in the form of a matrix representing the transformation of the bond length vector $d(C_{i},C_{i+1})$ into the NMR chemical shift vector $\delta_{i}$${}^{c}$ at linear order. For the special case of AnPixJg2 and Cph1Δ2, we used extensive QM/MM MD simulations and ab initio computations of ${}^{13}$C chemical shifts to determine the geometric sensitivity of the latter. We could show that with the help of multivariate regression this approach leads to a semi-quantitative understanding of the colour tuning mechanism, without the need to have the explicit molecular conformations of the chromophore.

A series of molecular conjugation scenarios using mesomeric formulae that explicitly describe the possibilities of bond alternation in PCB helps to connect the statistical predictions with the actual conjugation patterns found in P${}_{g}$ and P${}_{\mathrm{fr}}$ state. An important role is played by the stabilisation of the positive charge in tertiary carbocations or by ionic interaction with the protein.

Showcasing the examples of cyanobacteriochrome AnPixJg2 and phytochrome Cph1Δ2, our approach represents an alternative method to interpret NMR data that can readily be transferred to related tetrapyrrole chromophores in various protein environments.

## 5. Materials and Methods

### 5.1. Preparations

Starting geometries for P${}_{r}$ states of AnPixJg2 and Cph1Δ2 were obtained as X-ray structures from the Protein Data Bank (PDB; AnPixJg2: 3W2Z, [13] Cph1Δ2: 2VEA [35]). The simulation cell was set up using VMD, [74] solvating the protein in a box of TIP3P water [75]. Possible protonation states of protein residues were set to correspond to pH 7.0. The protonation state of histidine was chosen as follows: AnPixJg2: His293-E, His322-P, rest: D; Cph1Δ2: His260-E, His290-D, rest: D. The PCB chromophore was chosen with fully protonated pyrrole rings according to the experimental evidence [68].

### 5.2. Force Field Molecular Dynamics Simulations

FFMD simulations were carried out with the NAMD package [76] under periodic boundary conditions (PBC) using the CHARMM22 force field [77,78] in the isothermal-isobaric (NPT) ensemble and the combined Nosé–Hoover Langevin piston method with a period of 200 fs [79,80]. For PCB, force field parameters obtained by Mroginski and co-workers were used [58]. For van der Waals interactions a cutoff of 10 Å was employed. The initial simulation cell dimensions were chosen to include a water layer of 30 Å thickness. Bond lengths between heavy atoms and hydrogen atoms were held fixed using the SHAKE algorithm [81] with a time step of 2 fs. The system was equilibrated, first, by optimisation of the water cell, followed by heat up and equilibration keeping the protein positions fixed. Then, the protein structure was optimised with fixed water positions. Final equilibration for 200 ps was carried out after heating up the entire simulation cell to the desired temperature within 200 ps, followed by the production run (see Table 3 for more details).

### 5.3. QM/MM Molecular Dynamics Simulations

Snapshots from FFMD simulations were transferred into a QM/MM setup. AIMD simulations were performed as the Born-Oppenheimer type in the NVT ensemble using CP2K 5.1 [82]. The partitioning of the protein into QM and MM part, as well as employed capping atoms, is listed in Table 4. QM/MM bond interfaces were handled using an optimised capping potential introduced between for the bond between α and β carbon atoms [83,84]. The MM part was described by the CHARMM22 force field (see previous section). The size of the QM box was set to 30.0, 30.0, 30.0 Å. The BLYP functional [85,86] was employed together with Grimme’s dispersion correction (D3), ref. [87] using the GPW scheme [88], GTH pseudo-potentials, [89,90,91] a density cutoff of 320 Ry, and the TZVP-GTH basis set. The QM/MM interaction term was evaluated within the GEEP scheme presented by Laino and co-workers [92,93]. Equilibration runs of length 5–10 ps were carried out under massive thermostatting using Nosé–Hoover chains of length 5 at 330 K and a coupling constant of 10 fs [80,94]. For subsequent production runs, QM and MM subsystems were coupled to separate thermostat units with a coupling constant of 500 fs.

### 5.4. Calculation of Nuclear Shieldings

NMR calculations were performed in the same setup as AIMD simulations, but the size of the QM/MM part was chosen adaptively such that RMSE of chemical shifts were below 1% (cf. Figure A1). Water molecules within a range of 2.6 from the chromophore were included in the QM part as well. Snapshots were extracted from the AIMD trajectory every 200 fs. All atoms in the QM region were treated at an all-electron level using the Gaussian-augmented plane waves (GAPW) method of CP2K 5.1 [95] with density cutoff of 400 Ry and pcS-3 basis set (pcS-2 on oxygen and sulphur atoms) [96]. The size of the QM box was set to 35.0, 35.0, 35.0 Å. Gauge origin was treated using the IGAIM method of CP2K, [95] which is an adaption of the CSGT method based on Atoms in Molecules [97].

### 5.5. Statistical Analysis

Principal component analysis (PCA) and multivariate regression was carried out with R using the packages tidyverse, ade4, and GGally [98]. For PCA the centred, unscaled ${}^{13}$C chemical shifts of PCB atoms C${}_{13}$–C${}_{18}$ were used as variables to account for the importance of fluctuations in these features. PCA was not used for dimensionality reduction; though, consequently, the number of components equalled the number of input variables (6; cf. Figure 4).

Multivariate regression and prediction was carried out, inverting the correlation matrix of ${}^{13}$C shifts with C-C bond lengths in rings C and D, shown in Figure 3, based on the relationship
(1)$$\Delta \delta_{i}^{c}\propto M_{ij}\cdot \Delta d_{j}\;,$$

$\Delta \delta_{i}^{c}$ being the change in ${}^{13}$C chemical shift (features), $\Delta d_{j}$ the change in bond length (outcome/prediction), and $M_{ij}$ denoting elements of the linear correlation matrix (Figure 3); using the previously obtained principal components.

## 6. Other

Python-based MDAnalysis 1.0.0 [99,100] was used for handling protein topologies. Geometric analysis, QM/MM setups, as well as general post-processing of data was carried out using our own implementations in Python 3.8 [101,102,103,104] and numerical libraries NumPy 1.19.1 and SciPy 1.5.2 [105,106]. Plots were generated with Matplotlib 3.3.0 [107]. Molecular visualisations were created with VMD [74] using the Tachyon Ray Tracer [108] All structural formulae were created with 
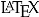
 using chemfig 1.56 [109].

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
