# Peer review of "Carbon Atoms Speaking Out: How the Geometric Sensitivity of 13C Chemical Shifts Leads to Understanding the Colour Tuning of Phycocyanobilin in Cph1 and AnPixJ"

_molecules, 2020, doi:10.3390/molecules25235505_

Round 1
Reviewer 1 Report
Comment on manuscript ID molecules-984338 „Carbon Atoms Speaking Out: How the Geometric Sensitivity of 13C Chemical Shifts Leads to Understanding the Colour Tuning of Phycocyanobilin in Cph1 and AnPixJ”
submitted to Molecules
The authors used advanced theoretical modeling involving GIAO NMR calculations in order to get an insight about the “Colour Tuning of Phycocyanobilin in Cph1 and AnPixJ”.
They presented a very detailed introduction to this important field. Thus, they performed “QM/MM study of PCB in protein using molecular dynamics (MD) and GIAO 13C-NMR chemical shielding/shift calculations. Thus, the single and double C-C bond changes are reflected in the calculated NMR chemical shifts. The size of NMR parameter variations are within few ppm and should be able to follow experimentally. They also analyse the mesomeric formulas of the studied molecular fragments. In particular, Gaussian-augmented plane waves (GAPW) were used for NMR calculations using Jensen basis sets, dedicated for NMR.
They used the dependence of the NMR chemical shifts on the bond length pattern what is very uncommon and shows a valuable novel approach to the problem.
I propose to accept the manuscript without changes and publish it in Molecules.
Author Response
We thank R1 for confirming relevance and impact of our work.
Reviewer 2 Report
The text by S. Jaehnigen and D. Sebastiani presents a
new method based on quantum and molecular mechanics
and chemical shift databases for deriving protein structure
information.
The method, though indirect, maximises information derived
from existing data and has real potential for investigations of other
proteins.
I recommend publication.
The several remarks are:
- the authors may wish to comment on the use of other data on
proteins, such as J-couplings (especially for dihedral angles -
separating between dihedral angle and chemical bond contributions
to fitted parameters is one aspect the study that deserves a more in-depth analysis), residual dipolar couplings,relaxation rate constants,
pseudo contact shifts, and on
how the method may be used on proiteins featuring paramagnetic
cofactors.
- the presentation is good, yet some phrases are long,
both in the main text and in the abstract. Some of the
terms are unusual, e.g., 'thusly' (pg 3): the commonly-used
counterpart of this adverb is simply 'thus'.
- the authors should point out whether in the source study
for chemical shifts (Matysik et al.) all protein measurements
are carried out at the same temperature, pH, etc.
Author Response
comment:
"the authors may wish to comment on the use of other data on proteins, such as J-couplings (especially for dihedral angles - separating between dihedral angle and chemical bond contributions to fitted parameters is one aspect the study that deserves a more in-depth analysis), residual dipolar couplings,relaxation rate constants, pseudo contact shifts, and on how the method may be used on proiteins featuring paramagnetic cofactors." (sic)
reply:
Clearly, there is a broader portfolio of experimental techniques (and their computational counterparts) which allow further investigation of structural data of proteins, notably J-couplings as mentioned by the reviewer. However, the focus of our work here has been the electronic conjugation of the _photoproduct_ state of the chromophore/protein. In such a (transient) state, it is not possible to obtain experimental J-couplings due to the short lifetime of the protein configuration and the resulting weaker signal intensity.
Also, while the J-couplings do give valuable information about covalent and non-covalent nuclear distances and angles, they are not correlated to electronic delocalization. A delocalized (conjugated) electron can "propagate" the nuclear spin/spin interaction equally well as a localized one. In contrast, the carbon NMR chemical shifts can indeed be linked to the electronic delocalization, which in turn is strongly correlated to the optical excitation (i.e. the absorption color, which is the primary interest of our study).
Hence, we have not looked on J-couplings in our present study. The same argument applies to the variety of complementary techniques mentions by the reviewer. Relaxation rates (diploar, quadrupolar) are primarily linked to the motional characteristics of the involved nuclei, but not the electronic structure.
comment:
"the presentation is good, yet some phrases are long, both in the main text and in the abstract. Some of the terms are unusual, e.g., 'thusly' (pg 3): the commonly-used counterpart of this adverb is simply 'thus'."
reply:
We thank R1 for the remarks, according to which we have simplified the convoluted passages.
comment:
"the authors should point out whether in the source study for chemical shifts (Matysik et al.) all protein measurements are carried out at the same temperature, pH, etc."
reply:
We have added several remarks to both, the computational details and the main text, stating that the chosen protonation state of the chromophore equals that of the experimental setup (4H) and that NMR shifts were recorded at 233 K. As mentioned in the text, we want to point out that temperature does not strongly affect 13C chemical shifts, which we supported by comparing theoretically obtained values (i.e., their RMSE) obtained likewise at 300 and 0 K (cf. Fig. A1).